# A Sliding Scale Signal Quality Metric of Photoplethysmography Applicable to Measuring Heart Rate across Clinical Contexts with Chest Mounting as a Case Study

**DOI:** 10.3390/s23073429

**Published:** 2023-03-24

**Authors:** Marnie K. McLean, R. Glenn Weaver, Abbi Lane, Michal T. Smith, Hannah Parker, Ben Stone, Jonas McAninch, David W. Matolak, Sarah Burkart, M. V. S. Chandrashekhar, Bridget Armstrong

**Affiliations:** 1Department of Exercise Science, University of South Carolina, Columbia, SC 29208, USA; 2College of Engineering and Computing, University of South Carolina, Columbia, SC 29208, USA

**Keywords:** photoplethysmography, PPG, heart rate, signal quality

## Abstract

Photoplethysmography (PPG) signal quality as a proxy for accuracy in heart rate (HR) measurement is useful in various public health contexts, ranging from short-term clinical diagnostics to free-living health behavior surveillance studies that inform public health policy. Each context has a different tolerance for acceptable signal quality, and it is reductive to expect a single threshold to meet the needs across all contexts. In this study, we propose two different metrics as sliding scales of PPG signal quality and assess their association with accuracy of HR measures compared to a ground truth electrocardiogram (ECG) measurement. Methods: We used two publicly available PPG datasets (BUT PPG and Troika) to test if our signal quality metrics could identify poor signal quality compared to gold standard visual inspection. To aid interpretation of the sliding scale metrics, we used ROC curves and Kappa values to calculate guideline cut points and evaluate agreement, respectively. We then used the Troika dataset and an original dataset of PPG data collected from the chest to examine the association between continuous metrics of signal quality and HR accuracy. PPG-based HR estimates were compared with reference HR estimates using the mean absolute error (MAE) and the root-mean-square error (RMSE). Point biserial correlations were used to examine the association between binary signal quality and HR error metrics (MAE and RMSE). Results: ROC analysis from the BUT PPG data revealed that the AUC was 0.758 (95% CI 0.624 to 0.892) for signal quality metrics of STD-width and 0.741 (95% CI 0.589 to 0.883) for self-consistency. There was a significant correlation between criterion poor signal quality and signal quality metrics in both Troika and originally collected data. Signal quality was highly correlated with HR accuracy (MAE and RMSE, respectively) between PPG and ground truth ECG. Conclusion: This proof-of-concept work demonstrates an effective approach for assessing signal quality and demonstrates the effect of poor signal quality on HR measurement. Our continuous signal quality metrics allow estimations of uncertainties in other emergent metrics, such as energy expenditure that relies on multiple independent biometrics. This open-source approach increases the availability and applicability of our work in public health settings.

## 1. Introduction

Accurate measurement of heart rate (HR) is crucial to inform health status metrics such as energy expenditure (EE) and chronic stress (i.e., heart rate variability). Abnormal HR patterns, such as elevated HR or low HR variability, are measurable manifestations of multisystem dysfunction that can be used to identify physiological responses to acute stress. This acute stress is then linked with unfavorable longer-term cardiometabolic outcomes [1,2]. Accurate HR measurement among free-living individuals is needed to advance the science of public health surveillance of factors related to chronic disease.

Electrocardiography (ECG) is a “gold standard” method of determining heart rate but is cumbersome in daily life settings, as it requires multiple leads that need to be changed daily. Photoplethysmography (PPG) is a convenient compact alternative technique for HR tracking. PPG works by recording blood volume changes using light reflection signals from the human body. PPG can be sampled at lower frequencies and therefore requires less energy than ECG, allowing longer battery life, an essential feature in wearable monitors [3]. An additional advantage of PPG over ECG is that it can be placed in any location where blood flow can be detected [4]. While many consumer wearable devices (i.e., Fitbit) have focused on collecting PPG signals from the wrist [5], alternative placements such as the chest may be preferable to reduce motion artifacts and reduce potential distraction [6].

PPG should be equivalent to ECG under ideal conditions [7,8] but appears to be inaccurate at high heart rates (>155) [9]. This is likely attributable in part to motion artifacts. Corruption of PPG signal by motion artifacts is a serious obstacle to the reliable use of PPG in ambulatory settings [10]. While removing motion artifacts from PPGs is critical, an essential first step is reliably detecting their presence, but few studies have focused on determining signal quality indices for PPG signals, especially in applied settings [11]. Currently, most methods focus on removal of motion artifacts, and there are few algorithms for simply detecting and quantifying PPG signal quality [12,13]. Breaking this step apart from artifact removal allows applied researchers more autonomy to decide which signals are deemed usable [14]. What applied researchers deem usable might vary by context. For example, PPG readings as part of a clinical stress test might require higher precision than a 14-day wear protocol for free-living individuals. Thus, a versatile signal quality indicator might have gradations of signal quality or a continuous quality score, which then allow researchers to tailor which signals are retained based on the precision needed. Reliable motion artifact detection techniques lay the foundation for a completely automated PPG data processing system that can identify PPG data frames contaminated with artifacts and further process them for motion artifact removal [10]. A first step in this process is identifying different degrees of poor signal quality and quantifying the impact of signal quality on accurate HR estimation. 

Even for signals that would be considered poor by current research standards, there is significant information present within the signal that could enable extraction of reliable HR. However, most PPG enhancement research has focused on motion artifact removal techniques. Only some have examined the ability to recover quasi-periodic information from data corrupted with motion artifacts [10]. Methods to recover data include wavelet analysis and decomposition techniques [15] and adaptive filters [16]. However, many signal quality and denoising techniques are computationally intensive [17] and require data from accelerometers [18], which may not be present in PPG devices. Thus, there is a need for computationally efficient signal quality detection techniques that use only PPG signal data. 

An additional limitation of existing signal quality and denoising literature is that most existing signal quality detection methods have only been tested on short time frame data (i.e., 2–3 s to 5 min) [19] and examined artificially (i.e., mechanical vibration) [20] or for minimal motion artifacts (i.e., finger tapping) [10]. Indeed, many algorithms are hard coded for specific data types and frequencies, which will likely limit their broadscale use [14] Even fewer algorithms have been developed and tested on original data [17]. Therefore, their application for measurement of free-living individuals is limited. As cited in several recent reviews in the literature [18,21], there is a need for the development of novel computationally efficient signal quality indicators that aid in the fidelity of vital parameter estimation, such as heart rate. Furthermore, there is a need to examine signal quality indicators from PPG signals collected from multiple different locations, including the chest. 

Although there is a wealth of scientific literature on validation studies of PPG and processing techniques [17,21] this work is rarely adopted by applied research fields, including public health. A potential reason for this limitation is the siloed nature of study teams [22]. Engineering teams are mainly concerned with signal synchronization, while sports medicine tends to focus on standardized protocols with targeted groups. Clinicians, on the other hand, tend to favor real-time remote telemonitoring applications. Alternatively, applied public health researchers need devices capable of monitoring free living HR over routine monitoring time frames (>7 days). A more diverse team science approach, involving engineers, exercise physiologists and applied public health researchers may provide a more robust approach to this design and processing problem. 

Therefore, the purpose of the current study is to describe a novel computationally efficient method to identify and quantify poor PPG signal quality. Metrics of signal quality are a critical first step to inform methods to recover signal information and ultimately produce accurate HR estimates. We use three separate datasets to calculate two continuous metrics of signal quality and examine the predictive value of the signal quality indices on HR measurement accuracy. Identifying signal quality falls within a larger method for a reduction of motion artifacts (ROMA) framework that aims to inform measurement of free-living HR outside the lab, where motion artifacts are a reality. The simplicity of the method has the potential to reduce computational time. The visual spectrogram analytics and open-source availability make this tool appealing to applied researchers, thereby overcoming a limitation of the field whereby advances in engineering are not readily adopted in applied public health research. This study is a unique contribution to the field for two main reasons: First, we aim to quantify signal quality and examine its impact on HR estimation, which is a foundational first step in motion artifact removal. Second, this study examines the signal quality of PPG signals collected from the chest, which may be a preferable location to collect PPG signal due to the reduction of motion artifacts and reduced distraction compared to wrist-based devices. The ability to detect poor PPG signal from chest-mounted PPG using open-source algorithms is a foundational first step toward designing novel open-source PPG devices that are ultimately adopted by health researchers to collect and process HR signal data from free-living individuals.

## 2. Materials and Methods

Three datasets were used in the current study, two existing datasets publicly available on PhysioNet [23] and one original dataset. The first dataset was the BUT PPG (Brno University of Technology Smartphone PPG) [24,25]. The BUT PPG is a dataset that contains a combination of clean PPG signals and PPG signals intentionally corrupted by motion artifacts. We used the BUT PPG dataset to determine if the ROMA metrics of signal quality were predictive of ground truth measures of signal quality. The second dataset we used was the TROIKA [26] dataset, which contains measures of HR using both PPG and ECG. The TROIKA dataset was used to determine if the metrics of signal quality were related to accuracy of PPG estimated HR (i.e., agreement between PPG HR and ECG HR). The third data set was original data collected by the study team at the University of South Carolina (UofSC). The third original dataset was used to examine the association between signal quality and HR accuracy from PPG collected from the chest. We used the original UofSC data to examine the associations between continuous ROMA metrics and HR accuracy and to examine the initial validity of cut points for ROMA signal quality metrics.

BUT PPG Dataset. The BUT PPG dataset [24,25] was created by the cardiology team at the Department of Biomedical Engineering, Brno University of Technology. It comprises 48 10 s recordings of PPGs and associated ECG signals used for determining reference HR. PPG data were collected by smartphone Xiaomi Mi9 (Xiaomi Inc., Beijing, China) with sampling frequency of 30 Hz. Reference ECG signals were recorded using a mobile ECG recorder (Bittium Faros 360, Bittium, Oulu, Finland) with a sampling frequency of 1000 Hz. Each PPG signal included annotation of quality and reference HR. Good and bad PPG signal quality was identified by expert visual inspection. PPG signal quality is rated using a binary criterion: 1 indicates good quality for HR estimation, 0 indicates signals where HR cannot be detected reliably, and thus these signals are unsuitable for any analysis. BUT PPG data were collected from 12 subjects (6 female, 6 male) aged between 21 to 61 years. Recordings were carried out between August 2020 and October 2020 [24,25].

TROIKA Dataset. The TROIKA dataset [26] consists of two-channel PPG signals collected on the wrist from individual trials from 12 male subjects between the ages of 18–35. Two pulse oximeters with green LEDs of wavelength 515 nm were embedded in a wristband, which was used to collect PPG signals sampled at 125 Hz. The ECG signal was recorded from the chest using wet ECG sensors. The 12 trials lasted a total of 5 min. Participants walked at 1–2 km/h for 0.5 min, then 6–8 km/h for 1 min, 12–15 km/h for 1 min, 6–8 km/h for 1 min, 12–15 km/h for 1 min and 1–2 km/h for the last 0.5 min. 

UofSC Dataset. The UofSC dataset consisted of 19 stationary bike sessions completed by 11 individuals. Laboratory generated data allowed us to have control over the sources of motion artifacts and the duration of activity to ensure that a wide variety of HRs were recorded. The study was conducted in accordance with the Declaration of Helsinki, and the study protocol was approved by the University of South Carolina IRB in August 2021 (Pro00107610). Informed consent was obtained from all subjects involved in the study prior to data collection. All data collection took place in the Clinical Exercise Research Center Lab at the University of South Carolina. Participants in the UofSC dataset were 11 healthy adults (Age 20–42) with no known history of cardiovascular disease or abnormalities. Participants completed between 1 and 4 trials on separate days for a total of 19 trials. Participants had between 2 and 6 on the Fitzpatrick skin tone scale [27,28].

UofSC Biking Protocol. For the laboratory dataset, the PPG sensor was worn on the chest, attached using a polyester spunlace adhesive [29,30]. The PPG sensor, which uses green light to measure HR, was purchased from PulseSensor.com. This vendor provides all part numbers and circuit board schematics, enabling open-source reproduction and traceability of device performance. The sensor was powered by an open-source Arduino board, which was also used to collect the PPG sensor response, enabling a time stamp for the measured data to be collected. This time stamp enabled synchronization with ECG telemetry (Polar H10 monitor (Polar, Singapore) described below) measurements to within 1 s, the minimum reported time segment. While this particular PPG sensor is continuously monitoring, the combination of the Arduino sampling and transmission/reception results in an effective received sampling rate of 46.3 Hz (sample period T*_s_* = 21.598 ms). This rate was determined using dummy time trials without a human subject to determine the number of samples received in 22 min, the length of the bike exercise protocol (including start up and stop times of one minute each). A Polar H10 chest strap heart rate ECG monitor was used as the comparison criterion (i.e., reference values) of HR. Polar monitors have been validated against the ECG gold standard [31].

All laboratory tests were performed indoors at 21 °C. For the protocol, subjects were asked to sit sedentary on the bike for the first 10 min to establish a consistent resting PPG signal. Then, subjects were asked to bike at a consistent speed of 50 RPM, which was monitored by an audible metronome. Participants biked at 50 RPM at moderate resistance for 2 min. For the next 3 min, the resistance was either increased by or maintained depending on the participant’s subjective exertion measured by the Borg perceived exertion scale [32]. After the 3 min, resistance was decreased, and participants were asked to rest for a final 5 min. For a total recording time of *T* = 20 min, the 46.3 Hz sampling rate yielded a data record of approximately *N* = 55,560 samples.

### 2.1. Signal Processing

The following steps were conducted for PPG signals from all three datasets (i.e., BUT PPG, TROIKA and UofSC biking data). The original sampled data are denoted by the sequence {*x_n_*}, for *n* = 1, 2…*N*. We also refer to sequences as vectors, via bold font, i.e., **x**. 

Preliminary motion artifact removal: The collected PPG data was processed in Matlab. The first step was to remove slow non-periodic motion artifacts that are inevitably present in all measurements arising from breathing, sweating, adhesive tension changes, etc. This slow baseline drift from non-periodic motion artifacts was isolated by performing a moving mean over 0.6 s to smooth out and suppress the systolic and diastolic peaks, typically <0.3 s in duration (Figure 1a), while preserving the other motion artifacts. Mathematically, we create sequence *y_n_* via convolution of **x** with a 0.6 s (length *L* = 30 sample) moving average (MA) filter with impulse response (IR) **h** (=ones(1,*L*)/*L*): **y** = **x** × **h**. The MA filter has the well-known Dirichlet frequency response *H(f) = sin(πLf/f_s_)/sin(πf/f_s_)*, which for our case has the first-null bandwidth of *f_s_*/30~1.543 Hz.

This baseline drift was then subtracted from the original signal to only show the systolic and diastolic peaks, which were now flat with respect to time, although their relative amplitude was not consistent over the trial (Figure 1b). The subtraction can be described as the creation of sequence **v** = **y*****q**, where **q** = *δ(n)* − **h**. Hence **v** is a high-pass filtered version of the original data **x**, *V*(*f*) = *X*(*f*)[1−*H*(*f*)]. This signal was then low-pass filtered at 3.5 Hz, i.e., 210 BPM to remove high frequency noise (e.g., from power lines), and smooth out the traces, while preserving the HR signal. The lowpass filter is a custom infinite IR (IIR) filter designed using Matlab’s “lowpass” function, with roll-off parameter specified by the steepness value of 0.8. At this point in the processing, we have sequence **z = v*h_lp_**, with **h_lp_** the IIR filter’s response.

The relative systolic peaks were then tracked using a moving maximum function (Figure 1b), producing sequence **w** = *movingmax*(**v**,0.5), where the functional operation employs a 0.5-s rectangular window. The HR amplitudes were then normalized, **r** = **w**/*max*(**w**), with the maximum taken over each 0.5-s interval; this yields HR signals that are the same amplitude (Figure 1c) to simplify beat counting in the time domain. 

Time Domain HR Measurement: For time-domain measurements of HR through systolic–systolic spacings, the HR amplitudes need to be normalized. As noted, this was accomplished using a moving maximum operation to detect the systolic peak amplitude over 0.5 s windows. Given that HR amplitude does not change appreciably over short times, this normalization was reliable over the different test subjects with varying HR, giving asymmetric PPG traces typically varying between about –0.5 and +1, although some negative excursions were larger. Once the HR was normalized, the systolic peak height and duration were identified using a 0.5 peak height threshold. For the subjects in our study, this did not cause spurious diastolic detection, as the diastolic peaks were typically close to 0 or negative. From the difference between subsequent beats, the instantaneous HR was determined: specifically, we computed a sequence of periods {T_0,*k*_}, where *k* was an index, and the *k*th period was *T*_0,*k*_ = *peak*(r_k_)−*peak*(r_k−1_), with the peak function selecting the maximum-valued sample within the *k*th 0.5 s window. This yields a sequence of HRs {HR*_k_*}, with *HR_k_* = 1/*T*_0,*k*_.

Outliers were removed from the HR sequence and replaced using Matlab’s “filloutliers” method, with a 40-beat moving median window, which removed points more than 3 local scaled MADs away from the local median. Outliers occurred through sharp jolts to the sensor due to poor mounting and will be described in the signal metrics section. These HRs were then averaged over 40 beats or ~30 s. The approximately 30 s window was chosen to be consistent with the 30 s time window used in the frequency domain determination of HR. While the average was over a 30 s window, a new HR in the time domain was computed for every beat, i.e., about once every second. A longer window smooths out variations due to signal noise from too short a window. This 30 s window also provides sufficient length to flag and average outliers in a robust fashion. Any outliers that were flagged were not used in the statistical calculations, with an estimated <0.1% of HR values being discarded. The discarded values were replaced with the value from the previous sample i.e., 21.6 ms before, too short a time for HR to change appreciably. Furthermore, for discarded values that were replaced, correlation with the frequency domain calculation using the full raw dataset described subsequently provided an additional check on accuracy. An example of poor quality PPG signal in the time domain is presented in Figure 2. 

Frequency Domain HR Measurement: A good example of a high-quality PPG signal with slow motion artifact free signal is shown in Figure 3a (without the subsequent low pass filter), after conversion to a spectrogram with a 30 s time window in Matlab leading to an HR value ~7.27 s using Matlab’s default windowing and overlap parameters. This time resolution is short enough to be clinically valid [33], while being long enough to capture multiple heartbeats for assigning a reliable frequency in the spectral/frequency domain. The spectrogram was computed for the sequence v defined previously, yielding *V*(*f*,*t*) over 30 s intervals. This magnitude-squared short-time Fourier transform allows estimation of peak power at the HR frequency over time. There is also power in the second harmonic as in all non-sinusoidal periodic signals [34], although the intensity is much weaker than the fundamental frequency at which the HR lies. The peak fundamental frequency powers were normalized to 1 so that the slow changes in the HR amplitude over the course of the trial (e.g., Figure 1b) did not distract from the key metric (HR or frequency). The peak power frequency and spectral line width (shown in yellow on Figure 3a–c) were determined at each time in the spectrogram and converted to an HR in BPM by multiplying by 60. The HR vs. time in the frequency domain was then interpolated back to the systolic time stamps determined in the time domain above for ease of comparison between the two domains. Figure 3a shows an example of an HR spectrogram with good quality signal along with the corresponding ECG telemetry HR and extracted frequency domain PPG HR overlaid. A poor signal from poor mounting, i.e., the sensor’s contact with the skin being broken and reformed leads to “streakiness” of the spectral line or broadband interference from the impulsive nature of the contact/re-contact effects, as shown in Figure 3b.

In some trials, a well-behaved periodic motion artifact arose at ~100 BPM, 2× the 50 RPM cadence during the pedaling phase on the bicycle. This artifact was removed using a custom IIR notch filter at 100 BPM with a 50 BPM width to account for variations in pedaling during the trial. This was also sufficient not to interfere with the actual HR signal. Before filtering, it was clear that there were two peaks in the spectrogram as determined by visual inspection, enabling complete recovery of the correct signal. The emergence of these weak motion artifacts could be an indicator of marginal mounting, although further investigation is needed to clarify this. For trials that gave a very poor match with the Polar H10 ECG telemetry (Figure 3c), the streaky signal did not give recoverable data. In other words, if the loss of contact with the skin was too severe, the distance between skin was so large that no signal corresponding to the HR was obtained. The “streakiness” in this case in the frequency domain was due to an abrupt change in the baseline PPG signal, i.e., a sharp voltage impulse, the FFT of which was nearly white-noise-like [34]. The signal processing steps are summarized in Figure 4. 

Reference Measure for Poor Signal Quality. In line with recent publications on motion artifact detection, we relied on expert human visual inspection to identify motion artifact corrupted data. Expert visual inspection is the current gold standard [19,35,36,37]. Visual inspection of UofSC data and TROIKA was conducted using spectrogram visualization plots (See Figure 3a–c, for example). The BUT PPG dataset included binary indicators of poor and good signal quality. Because the collected BUT PPG data records were only 10 s long, we used a sliding window of 2.5 s for frequency domain spectrogram calculation, which gives an HR value every 0.6 s using Matlab’s default windowing. While this short window length is not ideal for robust HR assignment, it was necessary due to the very short trials in the BUT PPG data.

ROMA Self-Consistency Signal Quality Index: Self-consistency (also known as HR frequency difference [38]) is defined as the difference between the fundamental frequency computed via the spectrogram *V*(*f*,*t*) and HR computed from the time domain peak calculation, the average *HR_k_*. This feature measures the agreement between the fundamental frequencies detected from the frequency spectrum and from the time-domain signal. It is assumed that the frequencies would be in agreement in a clean PPG segment. In a noise-corrupted segment, however, there could be large differences in the values. We computed the self-consistency metric as follows: The self-consistency between time domain and frequency domain is defined as the fraction of points that agreed to within 10 BPM, i.e., 1.94 times the 5 BPM limits of agreement of the threshold chosen.

ROMA Standard Deviation of Line Width (STD-Width) Signal Quality Index: Figure 3a,b show examples of spectrograms with good and bad agreement with the Polar telemetry reference values. In the “good” signal, the HR signal is sharp and well-defined in frequency, evidenced by the narrow yellow line in Figure 3a. As can be seen in Figure 3a, the width of this yellow spectral line does not change much, leading to a small standard deviation. In the “poor” signal (Figure 3b), the emergence of the interference streaks from loss of contact with the skin as described above leads to very wide yellow streaks when contact with skin is lost. Thus, the yellow line is dispersed throughout the spectrogram. When the abrupt change stabilizes, the yellow spectral line width (perhaps due to noise) becomes unpredictable until skin contact is re-established. This cyclical process leads to wide variations in the spectral line width due to poor mounting, giving a large standard deviation. Thus, the standard deviation in frequency of the spectrogram line is larger when there is poor signal quality, and the standard deviation of this line width is used as the ROMA STD-line with signal quality index. 

### 2.2. Statistical Analysis

We conducted two broad sets of analyses: In Part 1, we examined signal quality agreement. In Part 2, we examined the impact of signal quality on HR accuracy. 

Signal Quality Agreement: To assess signal quality agreement, for all 3 datasets we calculated point bi-serial correlation with 95% Bayes credible intervals (95% CI) between signal quality indicators (self-consistency and STD-width) and binary signal quality criterion values (good vs. bad per visual inspection). For the BUT PPG data, we also used ROC curves to identify area under the curve (AUC) and sensitivity/specificity for different values of self-consistency and STD-width compared with visual inspection signal quality (i.e., good/poor). We identified cut points that balanced both sensitivity and specificity, then applied them to the UofSC biking data. We used Kappa coefficients to examine agreement between signals identified as poor-quality determined using self-consistency and STD-width and gold standard visual spectrogram analysis. The Kappa statistic accounts for agreement expected by chance [39]. Kappa was interpreted based on the following scale described by Landis and Koch [40]: ≤0, poor agreement; 0.01–0.20, slight agreement; 0.21–0.40, fair agreement; 0.41–0.60, moderate agreement; 0.61–0.80, substantial agreement and 0.81–1.00, almost perfect agreement. We then conducted a binomial logistic regression to examine the unique and additive value of self-consistency and STD-width in predicting signal quality. 

Associations Between Signal Quality and Heart Rate Accuracy: To examine the impact of signal quality on HR accuracy, we calculated root mean square error (RMSE), mean absolute error (MAE) and mean absolute percent error MAE_(%)_ between calculated HR and ECG criterion heart rate. The formulas used for calculating these metrics are as follows:$$\mathrm{RMSE}=\sqrt{\frac{1}{n-1}\overset{n}{\underset{i=1}{\sum }}{\left(x_{i}-y_{i}\right)}^{2}}$$
$$\mathrm{MAE}=\frac{1}{n}\overset{n}{\underset{i=1}{\sum }}|x_{i}-y_{i}|$$
$${\mathrm{MAE}}_{\left(\%\right)}=\frac{1}{n}\overset{n}{\underset{i=1}{\sum }}|x_{i}-y_{i}|/x_{i}$$
where $x_{i}$ and $y_{i}$ are the respective PPG and Polar estimated HR at the *i*th aligned time point. We also report accuracy, defined as the percentage of points within 5 bpm of the criterion. We then conducted Pearson correlations with 95% Bayes Credible intervals (95% CI) to examine the association between RMSE, MAE and the ROMA signal quality metrics of self-consistency and STD width.

## 3. Results

### 3.1. Part 1—Signal Quality Agreement 

BUT PPG: Of the 48 observations in the BUT PPG dataset, 35 were marked as good quality. The remaining 13 were identified as poor-quality per the reference visual inspection criterion. Self-consistency was correlated with the binary signal quality indicator (r = 0.33, 95%CI 0.76 to 0.56) but STD-width was not (r = −0.15, 95%CI −0.41 to 0.12). ROC analysis from the BUT PPG data revealed that the AUC was 0.758 (95% CI 0.624 to 0.892) for STD-width and 0.741 (95% CI 0.589 to 0.883) for self-consistency. Based on optimal balance of sensitivity/specificity, we identified a cut off score of >30 for self-consistency (sensitivity = 0.615/specificity 0.80) and <10 for STD-width (sensitivity = 0.923; Specificity =0.571). 

Using the identified cut-off scores for STD-width < 10, 27 of the BUT PPG observations were identified as poor quality. Using the self-consistency cut off score > 30, 14 observations were identified as poor quality. A forward stepwise binary logistic regression model revealed that STD-width < 10 was a significant predictor of signal quality and explained 30% (Nagelkerke R^2^) of the variance in signal quality and correctly classified 73% of cases. Self-consistency > 30 did not add significant predictive value beyond STD-width and thus did not meet criteria to be entered in the logistic regression model. 

UofSC Biking Protocol: In the 19 sessions of the UofSC biking dataset, self-consistency for PPG signals ranged from 12.05 to 98.26, and STD-width ranged from 3.05 to 12.82 (see Table 1). There was a strong correlation between the visual inspection criterion of signal quality and self-consistency (r = 0.69, 95%CI 0.44 to 0.89) and STD-width (r = −0.64, 95% CI −0.87 to −0.36). Using the cut points identified above, 6 of 19 observations were identified as poor quality. There was substantial agreement [40] (Kappa = 0.872) between signals identified as poor quality using the criterion of visual spectrogram and both self-consistency > 30 and STD-width < 10 metrics. There was perfect collinearity (r = 1.00) between STD-width and self-consistency, thus logistic regressions could not be conducted. 

TROIKA: Of the 12 sessions, self-consistency for PPG signals ranged from 32 to 90, and STD-width ranged from 5.7 to 20.6. The criterion of visual spectrogram was correlated with both STD-width (r = −0.50, 95%CI −0.85 to −0.08) and self-consistency (r = 0.62, 95%CI 0.24 to 0.91). Using the identified cut points for STD-width < 10, 5 of the TROIKA observations were identified as poor quality. Using the self-consistency cut off score > 30, 0 observations were identified as poor quality. There was substantial agreement [40] (Kappa = 0.633) between signals identified as poor quality using visual spectrogram analysis and STD-width < 10. Because 100% of the signals were deemed high quality per the self-consistency metric, Kappa was not able to be calculated. 

### 3.2. Part 2—Associations between Signal Quality and Heart Rate Accuracy

TROIKA: The overall correlation between signal quality and HR accuracy between PPG and the ground truth of ECG (i.e., RMSE) was r = 0.56 (95%CI 0.17 to 0.90) for STD-width (see Figure 5a) and r = −0.38 (95%CI −0.81 to 0.08) for self-consistency (see Figure 5b). Similarly, MAE was positively correlated with STD-width r = 0.53 (95%CI 0.12 to 0.87) and negatively correlated with self-consistency r = −0.30 (95%CI −0.76 to 0.18).

UofSC Biking Data: Individual trial level accuracy, self-consistency, STD-width and errors (RMSE, MAE) are presented in Table 1. Aggregated averages stratified by signal quality are presented in Table 2. *T*-test revealed significant differences in terms of accuracy between protocols identified as poor (N = 5) and adequate signal quality (*n* = 14) using the binary cut points of self-consistency > 30 and STD-width < 10 (See Table 2). 

Signal quality was highly correlated with HR accuracy (MAE and RMSE, respectively) between PPG and the ground truth ECG Polar HR. Across all participants, the overall correlation between signal quality and HR accuracy between PPG and the ground truth of ECG (i.e., RMSE) was r = 0.77 (95%CI 0.57 to 0.92) for STD-width (see Figure 5a) and r = −0.73 (95%CI −0.91 to −0.51) for self-consistency (see Figure 5b). Similarly, MAE was positively correlated with STD-width r = 0.78 (95%CI 0.59 to 0.93) and negatively correlated with self-consistency r = −0.69 (95%CI −0.90 to −0.46).

Performance Comparison with Other Works: We compared our metrics of agreement (kappa) and association (correlation) with previous signal quality identification works which presented either kappa or correlation statistics (See Table 3). 

Neshitov et al. [41] also examined the TROIKA dataset for corrupted signal using wavelet transformation and found similar rates of poor quality signals ~40%. The current study and signal discarding ratio and self-consistency were highly correlated (r = −0.742).

**Table 3 sensors-23-03429-t003:** Signal quality identification: performance comparison with other works.

	Trials	Time (s)	N	Kappa	Correlation
Sukor [42]	104	60	13	0.64	-
Orphanidou [14]	1500	10	7	-	0.86
ROMA—UofSC Bike	19	1320	11	0.87	0.69
ROMA—Troika	12	300	1	0.63	0.62

## 4. Discussion

The purpose of this study was to describe a novel computationally efficient method to identify and quantify poor signal PPG quality. This is a necessary first step to recover signal information to produce accurate HR estimates. We demonstrated an effective method to identify poor signal PPG quality in both existing and original data and showed that signal quality is associated with HR accuracy. Identifying poor PPG signal is a critical first step before signal recovery methods can be used to ultimately produce accurate HR estimates. Both self-consistency and STD-width were associated with reference measures of signal quality. The new signal quality metrics were then associated with the accuracy of HR measured by PPG compared to an ECG in both existing and original data. Signal quality validity was evidenced by the strong correlation between signal quality and HR agreement between reference measures of HR (i.e., Polar telemetry and ECG) and PPG produced HR estimates. These findings indicate that poor signals are indeed producing inaccurate estimates of HR. While existing studies in this area suggest that, in short durations, PPG signals can produce accurate estimations of HR [19], this evidence is based on signals that were not collected from free-living individuals and included activities that have limited motion artifacts. These studies then have limited utility in applied research settings where motion artifacts are a reality. If advances in engineering and signal processing aim to have a public health impact, they need to overcome challenges including motion artifacts. 

Continuous measures of signal quality are needed to accurately distinguish valid HR measures in wearable devices. Current consumer wearable devices do not allow for open-source processing, and thus metrics are fundamentally unverifiable. This is especially worrisome in consumer wearables, which are some of the most used measures of PA in published studies, clinical trials and NIH-funded research [43]. However, similar concerns also exist among research-grade devices that use PPG, such as Empatica E4 and Biovotion Everion, given that the manufacturers prevent access to raw data. Thus, while devices will produce an HR estimate, the trustworthiness of that estimate is unknown. 

The open-source metrics of signal quality described in this study can be used in future PPG devices that aim to measure HR in free-living settings. Ideally, such devices should be capable of measuring multiple vital parameters, and this is an underdeveloped area according to Biswas et al. in their recent review [21] Further refinement should lead to the measurement of other hemodynamic markers through PPG, such as pulse wave velocity and augmentation index [44], both of which have high potential utility as health indicators [45,46]. Usage of these markers can provide a more feasible alternative to existing measures of blood pressure and pulse wave analysis, which require higher patient burden [44]. Additionally, further work needs to examine the effectiveness of the ROMA method in diverse populations across developmental stages and in settings that have ecologically relevant motion artifacts. It is worth noting that although we presented metrics of STD-width and self-consistency using a binary criterion of visual inspection (good vs. bad), statistically, it is usually preferable to work with the original continuous variables [47]. Indeed, using the continuous measure would allow applied researchers more flexibility over the minimal degree of signal quality deemed acceptable. Thus, while we present general guidelines for binary determination of signal quality, these are only intended to function as guidelines. Future research should aim to examine longer and more diverse PPG signals to examine the association between continuous signal quality and accuracy of HR measurement. 

Our study provides a computationally non-intensive method of estimating continuous signal quality from PPG collected from the chest. This is a foundational first step in the future of open-source signal processing. This finding also has high clinical utility for applied health researchers. Devices that collect PPG from the chest may be especially relevant for cardiac monitoring of children, as existing wrist-based wearable monitors may be uncomfortable or distracting for small children, especially in free-living conditions where children are asked to wear devices over multiple days. A downfall in the field is that advances in engineering are not readily adopted in public health research. Therefore, the next steps in this process are to use the metric to identify signal quality, remove signal noise and then recover usable data. From here, HR processing using the frequency domain can potentially salvage poor signal data. These metrics will inform the processing of data from a completely open-source wearable device designed to measure HR using chest-mounted PPG signal. 

Study results should be interpreted in the context of their limitations. While our sample size is consistent with the existing literature [21], we only included 11 individuals in our study. Although this sample provided thousands of data points, it is challenging to generalize and compare these results to the larger population. Our study sample comprised a relatively homogenous group, consisting of mostly healthy, active, White individuals. While the evidence regarding the impact of skin tone on PPG signal quality appears limited [48], the magnitude of this effect on the population level across health metrics is still unknown [49]. To overcome these limitations, we used two additional publicly available datasets to supplement our results. It is necessary for such research to be open-source and accessible to researchers across domains. We can improve the synergy between basic and applied scientific fields by developing and using open-source research-grade devices to gather raw signal data and then sharing that data publicly using services such as PhysioNet [23]. With more data available, the ROMA method to identify poor signals can be further validated in more diverse populations and age groups. 

## 5. Conclusions

Poor PPG signal appears to produce inaccurate estimates of HR. The approach developed in the current study allows for two continuous measures of signal quality, which can then be used to decide if functional information still exists in the signal, if measurements should be discarded or if the results can be interpreted with caution. The level of acceptable PPG signal quality may be dependent on the ultimate use of the device. Therefore, there is a need for a collaboration between engineering and public health researchers to continually develop and refine methods to measure and assess markers of individual and population level health. By creating a fully verifiable and easy to implement method of open-source processing, the scientific community can leverage team science and joint innovation across disciplines to ultimately improve measurements of HR which have applied utility in multiple settings, including medical contexts and public health.

## Figures and Tables

**Figure 1 sensors-23-03429-f001:**
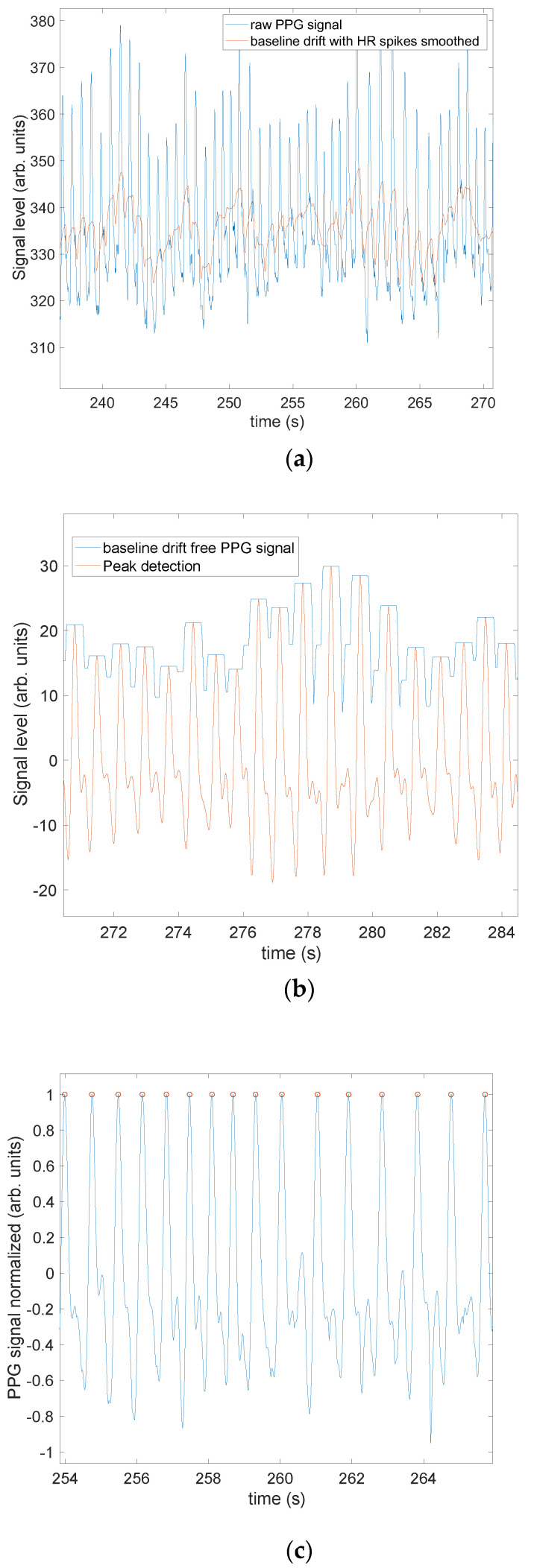
(**a**–**c**) Visualization of preliminary motion artifact removal. (**a**) Slow baseline drift from non-periodic motion artifacts isolated using a moving mean to smooth out the sharper systolic peaks that are to be isolated. This smooth baseline is then subtracted from the original signal so that only the desired HR signal remains. This is a time-domain high-pass filter. (**b**) Relative systolic peaks are tracked using a moving maximum function over 0.5 s. (**c**) Heart rate amplitudes are normalized using the amplitudes tracked in (**b**) so that all systolic features are of the same amplitude. Typical positive swings of +1 are typically mirrored by asymmetric negative swings ~0.5, enabling identification of each heartbeat. Given that the diastolic peaks are typically < 0, any sharp peak > 0.5 is identified as a systolic feature.

**Figure 2 sensors-23-03429-f002:**
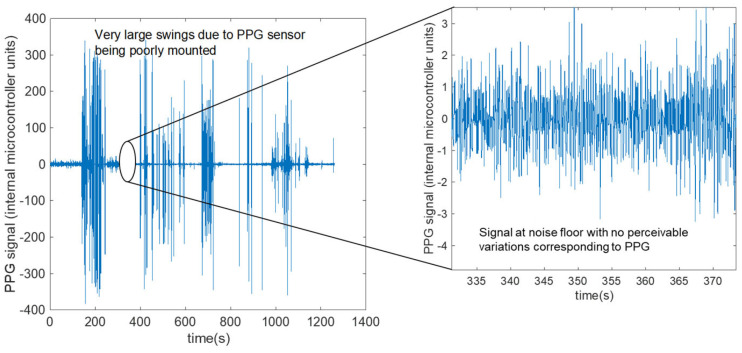
Poor quality PPG signal in time domain. The very large swings are from contact to skin being lost and reformed. These large swings are also responsible for the streaking seen in the spectrogram in Figure 3b. In other words, sharp impulses in time become broad streaks in frequency through the short time Fourier transform embedded in the spectrogram in Figure 3.

**Figure 3 sensors-23-03429-f003:**
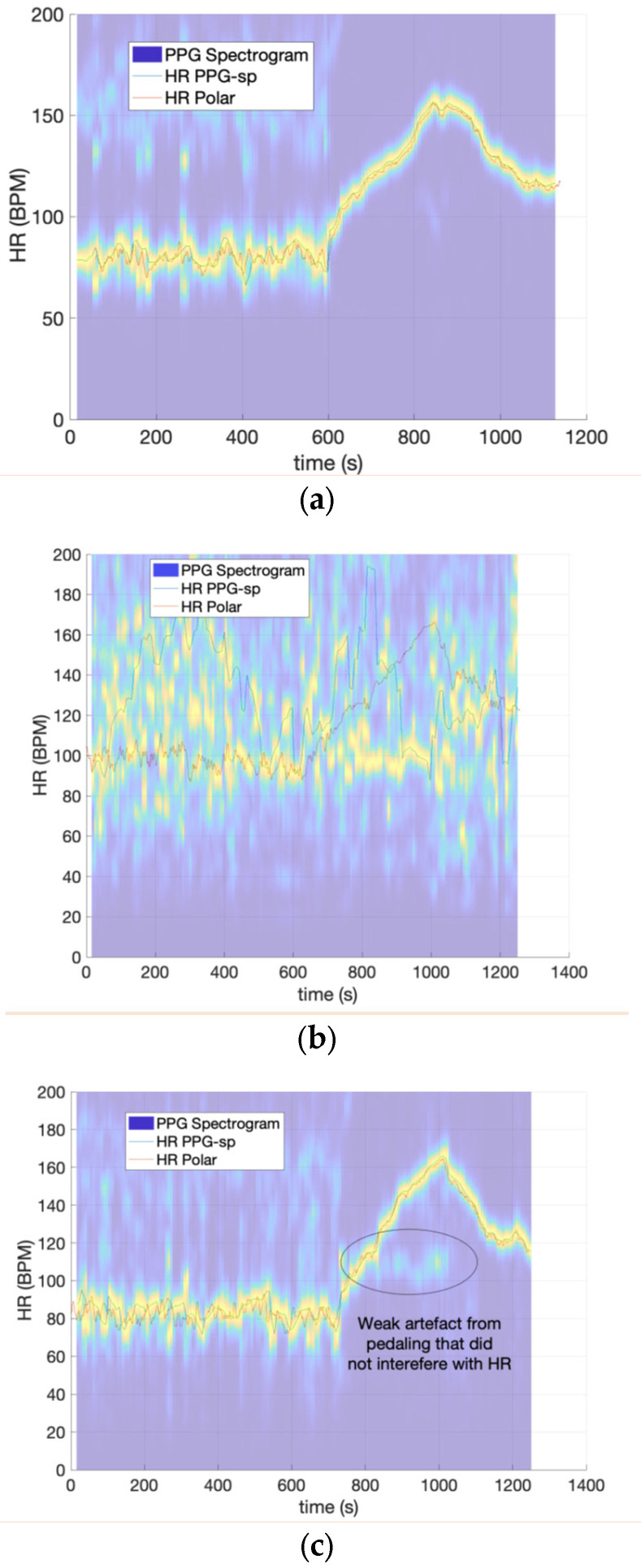
(**a**–**c**) Signal quality spectrograms. (**a**) Good signal quality spectrogram. (**b**) Poor signal quality spectrogram. (**c**) Good signal quality spectrogram with motion artifacts.

**Figure 4 sensors-23-03429-f004:**
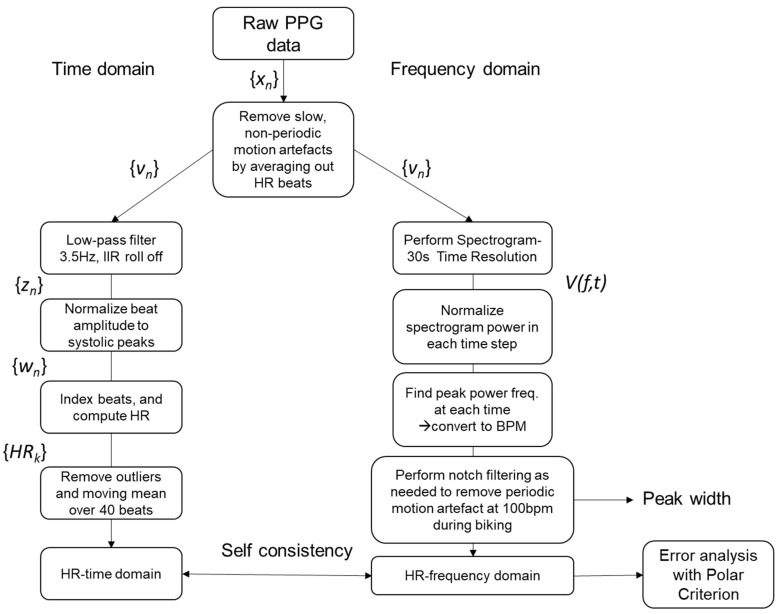
Heart rate processing system for PPG signal.

**Figure 5 sensors-23-03429-f005:**
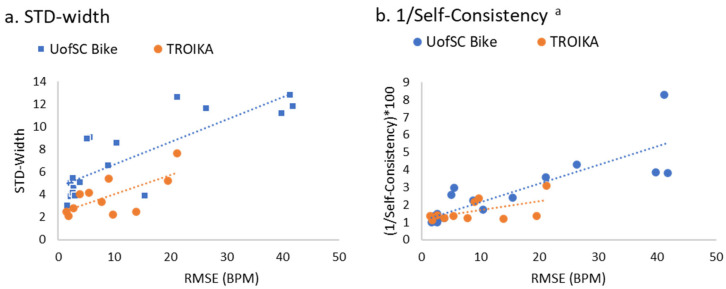
Association between PPG and ECG agreement (RMSE) and signal quality indices of STD-width (**a**) and self-consistency (**b**) in TROIKA and UofSC data. ^a^ Self-consistency is plotted as 1/self-consistency for ease of visual interpretation.

**Table 1 sensors-23-03429-t001:** Individual session signal quality, accuracy, RMSE and MAE metrics for UofSC Bike Data.

Session	ID	Accuracy	Self-Consistency	STD-Width	RMSE	MAE (BPM)	MAE _(%)_
1	2	14.38%	23.2	11.63	26.38	26.34	24.90%
2	3	12.38%	26.2	11.86	41.78	35.33	36.70%
3	3	15.43%	25.91	11.19	39.76	34.32	30.91%
4	1	93.13%	94.91	4.07	2.32	1.74	1.89%
5	1	96.57%	98.26	3.05	1.66	1.4	1.45%
6	1	93.74%	73.95	5.02	2.23	1.62	1.99%
7	1	93.22%	93.37	3.86	2.28	1.78	1.97%
8	3	75.72%	33.43	9.06	5.58	3.71	4.06%
9	9	90.54%	67.35	4.61	2.7	2.03	1.79%
10	5	14.48%	12.05	12.82	41.26	32.6	30.94%
11	2	62.83%	58.14	8.56	10.43	6.3	5.27%
12	6	91.43%	98.03	4.18	2.62	2.09	1.91%
13	7	87.86%	96.57	5.49	2.56	2.07	2.19%
14	10	34.59%	28.04	12.66	21.12	15.71	15.13%
15	13	69.51%	44.38	6.58	8.82	5.76	5.80%
16	7	79.02%	81.69	5.13	3.89	2.93	3.27%
17	10	78.20%	41.46	3.90	15.46	6.58	5.83%
18	11	77.73%	38.73	8.96	5.1	3.52	3.58%
19	14	89.67%	83.96	3.92	2.95	2.11	1.81%

RMSE—root mean square error; MAE—mean absolute error; STD—standard deviation; accuracy defined as the % of points within 5 bpm of the criterion.

**Table 2 sensors-23-03429-t002:** Accuracy of HR estimation by signal quality in UofSC bike data.

	Adequate Signal (N = 14)				Poor Signal (N = 5)				
	Mean	Min	Max	Std. Deviation	Mean	Min	Max	Std. Deviation	*p* < 0.01
Accuracy (%)	84.23	62.83	96.57	10.34	18.25	12.38	34.59	9.20	*
RMSE (BPM)	4.90	1.66	15.46	4.02	34.06	21.12	41.78	9.62	*
MAE (BPM)	3.12	1.40	6.58	1.82	28.86	15.71	35.33	8.14	*
MAE (%)	3.06	1.45	5.83	1.59	27.72	15.13	36.70	8.18	*

* *p* < 0.01 between poor and adequate signal quality.

## Data Availability

Data and corresponding processing code are publicly available on PhysoNet at Github, respectively https://github.com/ACOI-UofSC/Bike_Protocol.

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
