# Peer review of "A Sliding Scale Signal Quality Metric of Photoplethysmography Applicable to Measuring Heart Rate across Clinical Contexts with Chest Mounting as a Case Study"

_sensors, 2023, doi:10.3390/s23073429_

Round 1

Reviewer 1 Report

1. The object of signal quality evaluation is the entire data, and the evaluation of signal quality is based on the statistical results of the signal quality evaluation indicators obtained every 30s of the entire data. Dose the signal length affect the evaluation results? On the other hand, by evaluating the entire piece of data, much useful information may be lost considering the way poor quality signals are handled (such as discarding). Please provide the rationale for calculating a heart rate every 30s.

2. In the algorithm, the author used a 30s time window for heart rate calculation, but the first dataset (BUT PPG dataset) is only 10s long, how to calculate the signal quality evaluation index? In addition, no time information about the TROIKA Dataset was provided.

3. The second indicator (ROMA Standard Deviation of line width (STD-width) signal quality index) is not clearly defined, and what HR signal line width refers to needs to be discussed in more detail.

4. One purpose of signal quality evaluation is to obtain more accurate heart rate, but the author did not analyze how to obtain a more accurate heart rate through signal quality assessment.

Author Response

We appreciate the reviewer’s time and efforts on this manuscript – below please find the responses to reviewer comments. Changes to the manuscript are highlighted in orange in the main document.

  1. The object of signal quality evaluation is the entire data, and the evaluation of signal quality is based on the statistical results of the signal quality evaluation indicators obtained every 30s of the entire data. Dose the signal length affect the evaluation results? On the other hand, by evaluating the entire piece of data, much useful information may be lost considering the way poor quality signals are handled (such as discarding). Please provide the rationale for calculating a heart rate every 30s.

Response: We updated the methods section to clarify and provide rational on this point based on our understanding of the reviewer’s comment. Briefly, we calculated HR every ~7 seconds, but used a 30 second moving window to compute the spectrogram. While the average is over a 30s window, a new HR in the time domain is computed for every beat (i.e. about once every second). A longer window smooths out variations due to signal noise from too short a window. This 30s window also provides sufficient length to flag and average outliers in a robust fashion. Any outliers that were flagged are not used in the calculation, with an estimated <0.1% of HR values being discarded. The discarded values are replaced with the value from the previous sample i.e. 21.6ms before, too short a time for HR to change appreciably. Furthermore, for discarded values that were replaced, correlation with the frequency domain calculation using the full raw dataset described below provides an additional check on accuracy. (See p. 6-7)

  1. In the algorithm, the author used a 30s time window for heart rate calculation, but the first dataset (BUT PPG dataset) is only 10s long, how to calculate the signal quality evaluation index? In addition, no time information about the TROIKA Dataset was provided.

Response: We now clarify this point in the methods section. Because BUT PPG data was collected only 10 seconds long, we used a sliding window of 2.5 seconds for frequency domain spectrogram calculation, which gives a HR value every 0.6s using Matlab’s default windowing. While this short window is not ideal for robust HR assignment, it was necessary due to the very short trials in the BUT PPG data (See p. 8).  We also now include information about the total time Troika trials lasted (5 minutes) (see p. 4).

  1. The second indicator (ROMA Standard Deviation of line width (STD-width) signal quality index) is not clearly defined, and what HR signal line width refers to needs to be discussed in more detail.

Response: We presume that by ‘HR signal line’ the reviewer is referring to the spectral signal line derived from the frequency domain spectrogram. The spectral line corresponds to the identified fundamental frequency (i.e., peak power frequency) over time. We now clarify that the standard deviation of this spectral line (i.e., the identified fundamental frequency across time) is what we use as the signal quality metric ROMA STD-width. This clarification can be found under the “ROMA Standard Deviation of line width (STD-width) signal quality indexheading on p. 9 and on p. 7 when describing the process of frequency domain signal processing.

  1. One purpose of signal quality evaluation is to obtain more accurate heart rate, but the author did not analyze how to obtain a more accurate heart rate through signal quality assessment.

Response: The reviewer is correct that we present only a metric of signal quality, which is not intended to improve HR accuracy in itself.  Rather, we intend to show that our metric signal quality is related to accuracy and can be considered on a sliding scale. Depending on the applied context, this scale could be used to determine the appropriateness of retaining or discarding data in a separate subsequent step. We now further clarify this point in the introduction and discussion (see p. 13).   

Reviewer 2 Report

In this paper, the authors proposed 2 different metrics as sliding scales of PPG signal quality and assess their accuracy to measure HR as compared to a ground 16 truth electrocardiogram measurement. Although the topic and the method are interesting, there are some points should be improved. They are given below.

The Troika dataset should be elaborately introduced in terms of number of recorded PPG trials.

The metrics should be mathematically introduced.

How the HR accuracy is calculated. It should be mathematically introduced.

Some poor and quality PPG signals should be shown with figures.

The abbreviation CI must be given with long version at the first usage.  

How can the authors compare their results and show their effectiveness by comparing them with the previously reports studies?

Author Response

We appreciate the reviewer’s time and efforts on this manuscript – below please find the responses to reviewer comments. Changes to the manuscript are highlighted in orange in the main document.

  1. In this paper, the authors proposed 2 different metrics as sliding scales of PPG signal quality and assess their accuracy to measure HR as compared to a ground 16 truth electrocardiogram measurement. Although the topic and the method are interesting, there are some points should be improved. They are given below.

Response: We appreciate the reviewer’s interest in this topic.

  1. The Troika dataset should be elaborately introduced in terms of number of recorded PPG trials.

Response: We now describe the number, time and protocol requirements for the Troika dataset (see p. 4).

  1. The metrics should be mathematically introduced.

Response: We appreciate the reviewer’s suggestion on this matter, and we now introduce the signal processing mathematically in the materials and methods section (see p. 5-8) and link this to the steps in Figure 4.

  1. How HR accuracy is calculated. It should be mathematically introduced.

Response: We now include the formulas to calculate the metrics of accuracy including RMSE, MAE and MAE% (see p. 9-10). We also report that ‘accuracy’ is defined as the % of points within 5 bpm of the criterion both in the table 2 footnote and in the statistical analysis sections.

  1. Some poor and quality PPG signals should be shown with figures.

Response: We now include additional figures demonstrating poor quality signals in both the frequency and time domain (see Figure 2a-d)

  1. The abbreviation CI must be given with long version at the first usage.  

Response: We now introduce the 95% credible interval for the abbreviation CI (see p. 9)

  1. How can the authors compare their results and show their effectiveness by comparing them with the previously reports studies?

Response: We now include a table in the result comparing our signal quality identification results with other works using different datasets.  We also report on the correlation between our work previous studies identifying signal quality using the TROIKA dataset (see p. 13 & Table 3).

Round 2

Reviewer 2 Report

The authors addressed all the points and now it can be published in the journal.